# Invasion and high-elevation acclimation of the red imported fire ant, *Solenopsis invicta*, in the southern Blue Ridge Escarpment region of North America

A. J. Lytle[1¤]*, J. T. Costa[1,2], R. J. Warren, II[3]

1 Department of Biology, Western Carolina University, Cullowhee, North Carolina, United States of America, 2 Highlands Biological Station, Highlands, North Carolina, United States of America, 3 Department of Biology, SUNY Buffalo State, Buffalo, New York, United States of America

¤ Current address: North Carolina State University, Raleigh, North Carolina, United States of America
* ajlaffe2@ncsu.edu

**Data Availability Statement:** All data files are available from the Dryad database: https://datadryad.org/stash/share/DeFOMcoOqf5Lgtu7KmDOHLywfIZGG6nLJzeYiPE3_0o.

## Abstract

The red imported fire ant (*Solenopsis invicta*) is a non-native invasive species that rapidly spread northward in the United States after its introduction from South America in the 1930s. Researchers predicted that the northward spread of this invasive ant would be limited by cold temperatures with increased latitude and greater elevation in the Blue Ridge Escarpment region of the United States. The presence of *S. invicta* at relatively high elevations north of their projected limits suggests greater cold tolerance than previously predicted; however, these populations might be ephemeral indications of strong dispersal abilities. In this study, we investigated potential physiological adaptations of *S. invicta* that would indicate acclimation to high elevation environments. We hypothesized that if *S. invicta* colonies can persist in colder climates than where they originated, we would find gradients in S. *invicta* worker cold tolerance along a montane elevational gradient. We also predicted that higher elevation *S. invicta* ants might incur greater physiological costs to persist in the colder climate, so we measured colony lipid content to assess health status. For comparison, we also collected physiological temperature tolerance data for the co-occurring dominant native woodland ant *Aphaenogaster picea*. We found that *S. invicta* occurring at higher elevations exhibited greater physiological tolerance for cold temperatures as compared to lower-elevation conspecifics–a cold tolerance pattern that paralleled of the native *A. picea* ants along the same gradient. Both *S. invicta* and *A. picea* similarly exhibited lower thermal tolerances for colder temperatures when moving up the elevational gradient, with *A. picea* consistently exhibiting a lower thermal tolerance overall. There was no change in *S. invicta* colony lipid content with elevation, suggesting that greater metabolic rates were not needed to sustain these ants at high elevations.

## Introduction

Researchers have long believed that introduced *Solenopsis invicta* [1] populations in the southeastern United States would not expand to higher latitudes and elevations because their pre-

**Funding:** Amanda Lafferty (A.J.L) received funding from a Grant-in-Aid of Research and Ralph Sargent Scholarship provided by the Highlands Biological Foundation, Inc (Grant number HBS-GIA-2017-05). The funders had no role in study design, data collection and analysis, decision to publish, or preparation of the manuscript.

**Competing interests:** Amanda Lafferty (A.J.L) received funding from a Grant-in-Aid of Research and Ralph Sargent Scholarship provided by the Highlands Biological Foundation, Inc (Grant number HBS-GIA-2017-05). This does not alter our adherence to PLOS ONE policies on sharing data and materials. Highlands Biological Station URL: https://highlandsbiological.org/.

adaptation to the subtropical climate of their native range would make them unable to survive prolonged cold weather exposure [2–4]. For example, Korzukhin et al. [2] predicted that winter temperatures would limit *S. invicta* alate production through freeze-kills and stunted reproductive output, and therefore the ant would be unlikely to colonize areas with minimum temperatures below -3.7 °C. As such, the southern Blue Ridge Escarpment region in western North Carolina, U.S. (the zone of abrupt change in elevation between the Blue Ridge and Piedmont physiographic provinces with a vertical relief of 400 m to ca. 760 m) was projected as unsuitable habitat for *S. invicta* due to the cold temperature extremes at high elevations [2]. Within the last five years at least, however, *S. invicta* colonies have been observed at elevations > 1220 m in the Blue Ridge Escarpment where temperatures reached anomalous minima of -16.3 °C in Macon County, North Carolina, in 2019 [5, 6].

The objective of our study was to explore potential mechanisms that might explain the persistence of *S. invicta* colonies through the winter season and high elevation acclimation. We investigated *S. invicta* physiological thermal tolerance along an elevation gradient to determine whether the ants were able to physiologically acclimate to colder temperatures. For comparison, we also tested native *Aphaenogaster picea* [7] thermal tolerance along the same gradients. We hypothesized that *S. invicta* would have a lower critical thermal minimum ($CT_{min}$) at higher elevations and that the shift in thermal tolerance would be similar to that of the native ant, *A. picea*. We also predicted that *S. invicta* critical thermal maximum ($CT_{max}$) would be higher than *A. picea* ants because *S. invicta* originates from a subtropical climate and occurs in open habitats (in contrast to the forest-dwelling *A. picea* ants).

The lipid content of ant colonies gives an indication of colony health and can be useful in predicting the reproductive success of a colony [3, 8, 9]. We expected that *S. invicta* colonies at higher elevations (915 m and above) would be less healthy (and hence have a lower lipid content) due to the potential shorter optimal foraging time and because they may have to expend more metabolic energy to persist in colder temperatures.

## Methods

### Study species

*Solenopsis invicta* is one of several fire ant species native to South America. Specifically, *S. invicta* is indigenous to southern Brazil, Paraguay, and northeast Argentina, and has been introduced to many countries over the past two centuries, including the U.S. where it has rapidly spread since its first introduction to Mobile, Alabama, in the 1930s [10]. Habitat disturbance, a perturbation that is caused by either biotic or abiotic forces [11], such as the construction of buildings and roads or the removal of biomass by a natural force is crucial for the establishment of many invasive species including *S. invicta*. Globalization and the increase of trade has allowed this species to spread successfully to many disturbed environments via shipping and other modes of transport [12].

*Aphaenogaster picea* ants are native to deciduous forests in the eastern U.S. and are not only one of the most abundant ants in this region, but they are important ecologically as seed dispersers of many native understory herbs such as *Sanguinaria*, *Trillium*, and *Hepatica* spp. [13, 14]. *Aphaenogaster picea* can coexist in the same general area as *S. invicta*, however, *S. invicta* primarily inhabits disturbed, full-sun environments whereas *A. picea* inhabits shady forests [3, 14].

### Collection sites

*Solenopsis invicta* workers were collected at three elevational ranges (hereafter, "sites"): low (0–305 m), mid (457–762 m), and high (> 915 m) [Fig 1]. The highest elevation population of *S. invicta* was collected at 1228 m on a logging trail in Macon County, North Carolina. The

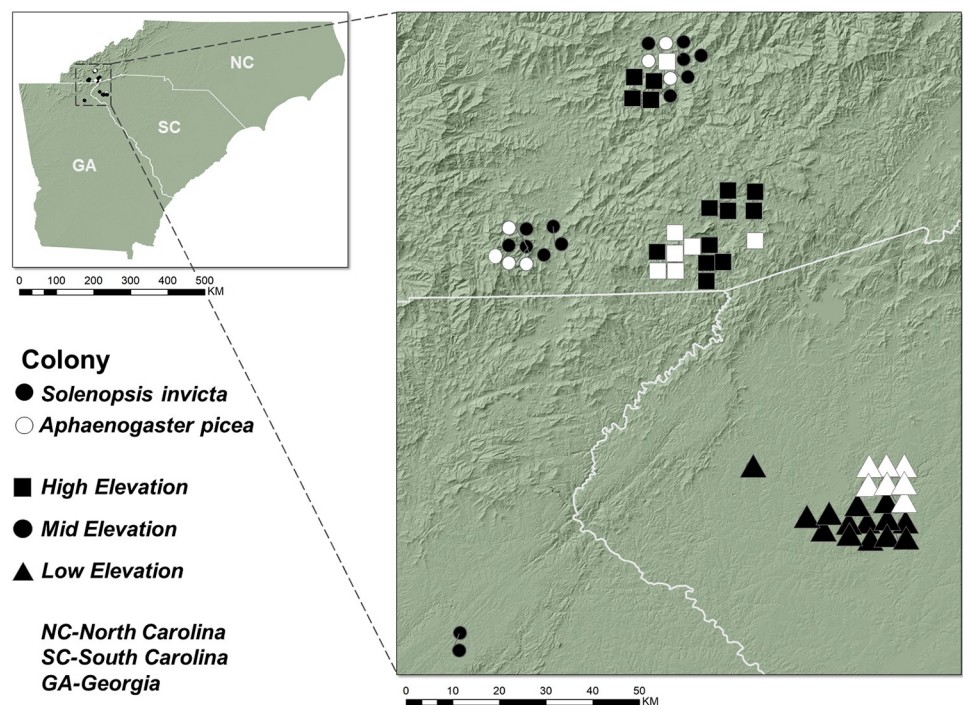

**Fig 1. Map of North Carolina, South Carolina, and Georgia, U.S., where all ant colonies were collected.** Reprinted from National Boundaries Dataset, 3DEP Elevation Program (https://www.usgs.gov/core-science-systems/ngp/3dep) with permission from USGS, public domain 2017, as well as from the 2017 TIGER/Line Shapefiles (https://www.census.gov/geo/maps-data/data/tiger-line.html) with permission from the U.S. Census Bureau, public domain 2017.

lowest-elevation *S. invicta* population collected was at 203 m elevation in Clemson, South Carolina (spanning 1025 m). The "low elevation" collecting locations included *S. invicta* colonies in Anderson County and Oconee County, South Carolina, with the majority collected along Tiger Boulevard in Clemson, South Carolina (N34.692611 W-82.847204). The "mid-elevation" collecting locations included colonies in Rabun County, Georgia, and Macon and Jackson Counties, North Carolina such as the Western Carolina University campus (N35.309325 W-83.18485) and Coweeta Hydrologic Laboratory (N35.059707 W-83.430739). The "high elevation" collecting locations included colonies in Macon and Jackson counties, North Carolina on or near the southern Blue Ridge Escarpment such as the Lonesome Valley residential community in Cashiers, North Carolina (N35.131682 W-83.062290), the Cashiers Recreation Center (N35.110823 W-83.104809), the Chattooga Narrows hiking trail (N 35.040692 W -83.136446), and a private logging road in Franklin, NC (N35.279735 W-83.231491).

We found *S. invicta* colonies abundant in disturbed areas along road rights-of-way, highway rest stops, and along the edges of agricultural fields. All the *S. invicta* colony that we observed and collected were established in anthropogenically disturbed habitats, and we did not find any colonies in undisturbed habitats (personal observation). For study sampling, we collected from active, mature colonies that were at least ten meters from other colonies. If we failed to find a colony after fifteen minutes of searching, we moved to another location. We collected 300 worker ants from fourteen *S. invicta* colonies at each of the three sites ($n = 42$ colonies) occurring along a piedmont-to-mountain elevational gradient. Additionally, fifteen workers from seven *A. picea* colonies were collected at each of the three sites ($n = 21$ colonies) as close to the *S. invicta* colonies as possible. All ants were collected in June and July 2017. The

GPS location of each ant colony sampled was recorded with a GPS unit (Garmin GPSMAP 64s) and recorded for mapping purposes through ArcMap (ESRI, Inc., Redlands, CA). No permits were required for field collection; we obtained permission to collect ants from private residential areas and the remainder of the ants were collected from public spaces. *Solenopsis invicta* and *A. picea* ants from each colony were placed in plastic bags in a cooler and promptly brought back to the Highlands Biological Station laboratory (Macon Co., North Carolina) and stored in a refrigerator until thermal tolerance tests were performed, no more than 24 hours after field collection. Following the thermal tolerance assays, the rest of the collected *S. invicta* ants were freeze-killed in a -80 °C freezer to use in the lipid analysis tests.

## Thermal tolerance

Lower and upper thermal tolerance tests were performed to determine physiological limits for *S. invicta* and *A. picea*. We tested 30 ants from each of the 42 *S. invicta* colonies and 15 ants from each of the 21 *A. picea* colonies. *Solenopsis invicta* are polymorphic, so we separated the larger "major" and smaller "minor" workers of *S. invicta* colonies for the assay. We tested equal numbers of *S. invicta* majors ($n = 15$) and minors ($n = 15$), whereas *A. picea* only has one worker size, so we tested 15 total workers from each colony. All workers were placed individually in 16 mm labeled glass test tubes with moistened filter paper to prevent desiccation and plugged with cotton to prevent escape. Ten ants from each *S. invicta* colony (5 majors and 5 minors) were kept in control test tubes at room temperature (~24 °C) for the duration of the thermal testing trials. Five minors and five majors from each *S. invicta* colony were used for the cold tolerance test. They were placed in the individual test tubes in a Thermo Fisher ARCTIC A40 refrigerated water bath (NesLab, ThermoScientific, Portsmouth, NH, USA) and acclimated for ten minutes at 20 °C. Temperatures were then decreased by 1°C min$^{-1}$. One minute after the water bath reached the desired temperature, the ants were observed. If an ant was moving, it was placed back into the water bath where it remained for another decrease of 1°C. $CT_{min}$ was measured as the temperature at which the ants displayed a loss of motor control and were unable to right themselves after being flipped on their back. The temperature at which ants could no longer right themselves was recorded and represented the critical thermal minimum limit. A different set of ants from each colony was then tested for their $CT_{max}$ limits following similar methodology as $CT_{min}$ except that temperatures were stepped up from 30 °C. These methods were repeated for the *A. picea* thermal tolerance assays with five ants for each of $CT_{min}$, $CT_{max}$ and control.

## Lipid analysis

Following field collection, we freeze-killed *S. invicta* by storing them in a -80 °C freezer for at least three days. Two hundred ants from each colony were haphazardly selected from the samples and dried at 60 °C for 48 hours. The ants were weighed and their dry mass recorded. Lipids were removed using Soxhlet extraction following the protocol of Smith and Tschinkel [15] except that we sampled whole-colony lipids rather than individual ants. We filled a Soxhlet thimble with the entire sample of ants from each colony to get an estimate of whole-colony lipid content. After lipid removal, we dried the ants at 60 °C for 48 hours and recorded the weight.

## Data analysis

Differences in temperature limits between size classes (minors and majors) of *S. invicta* were tested with a Student's two sample t-test. We evaluated thermal tolerance ($CT_{min}$ and $CT_{max}$) as a function of species (*S. invicta* and *A. picea*) and elevation using multiple regression models. We included a species x elevation interaction term to test whether ant species thermal

response differed by elevation. We analyzed *S. invicta* colony lipid content as a function of elevation with a linear regression. Residual plots were used to verify that the assumptions of normality and homogeneity of variance were correct for all analyses. All statistical analyses were performed with the R statistical program "R Studio" [16].

## Results

### Thermal tolerance

There was no difference in thermal tolerance temperatures between *S. invicta* minor and major worker ants for $CT_{max}$ (*df* = 417, *t-value* = 1.610, *p-value* = 0.108) or $CT_{min}$ (*df* = 418, *t-value* = 1.373, *p-value* = 0.171). The mean (±SE) $CT_{max}$ was 46.5 ± 0.1˚C for major workers and 46.3 ± 0.1˚C for minor workers. The mean $CT_{min}$ was 6.9 ± 0.2˚C for major workers and 6.6 ± 0.2˚C for minor workers. Therefore, we did not separate the two size classes of worker ants in further analyses.

*Aphaenogaster picea* tolerated lower minimum temperatures (5.2 ± 0.2˚C) than *S. invicta* (6.7 ± 0.1˚C) [Fig 2A; *coef.* = -1.321, *SE* = 0.318, *t-value* = -4.150, *p-value* < 0.001] whereas *S. invicta* tolerated higher $CT_{max}$ (46.4 ± 0.1˚C) than *A. picea* (43.2 ± 0.1˚C) [Fig 2B; *coef.* = -3.180, *SE* = 0.15, *t-value* = -21.19, *p-value* < 0.001].

There was no species x elevation interaction for $CT_{min}$ (*coef.* = -0.001, *SE* = 0.001, *t-value* = -0.723, *p-value* = 0.472) or $CT_{max}$ (*coef.* < -0.001, *SE* < 0.001, *t-value* = -0.528, *p-value* = 0.599), indicating that the thermal tolerances for both species decreased similarly with increased elevation. Minimum temperature tolerance decreased for both ant species with higher elevation (Fig 3A; *coef.* = -0.003, *SE* < 0.001, *t-value* = -8.443, *p-value* < 0.001). The $CT_{min}$ decreased by a mean of 2.8 ± 0.2˚C for *S. invicta* and 3.5 ± 0.4˚C for *A. picea* as elevations increased. Maximum temperature tolerance also decreased with elevation for both species (Fig 3B; *coef.* = -0.001, *SE* < 0.001, *t-value* = -3.627, *p-value* < 0.001) by a mean of 0.7 ± 0.2˚C for *S. invicta* and 0.8 ± 0.2˚C for *A. picea*.

### Lipid analysis

*Solenopsis invicta* colony lipid content did not change with elevation (S1 Fig; *coef.* < 0.001, *SE* < 0.001 *t-value* = -1.925, *p-value* = 0.684). Lipid content for low elevation *S. invicta* colonies was 0.042 ± 0.021 g ant$^{-1}$, for mid elevation colonies it was 0.034 ± 0.022 g ant$^{-1}$, and for high elevation was 0.044 ± 0.014 g ant$^{-1}$.

## Discussion

*Solenopsis invicta* has indeed acclimated or adapted to cold temperatures at high elevations in the southern Blue Ridge Escarpment region. Both the cold and heat tolerance thresholds of *S. invicta* decreased with increasing elevation indicating a physiological ability to tolerate colder temperatures, and the shift in thermal tolerance paralleled that of the native ant *A. picea*. *Solenopsis invicta* lipid content did not vary with elevation, suggesting no metabolic effects of cold acclimation for the ants. These results suggest that *S. invicta* is not as limited by cold temperatures as previously thought and will likely continue to invade higher elevations and latitudes in the southeastern U.S.

The decrease in *S. invicta* $CT_{min}$ with higher elevation is consistent with other ants across elevation and heat gradients [17–20]. For example, Bishop et al. [18] found that African ant $CT_{min}$ decreased with decreasing temperatures along a 1500 m elevation gradient. Ant foraging phenology corresponded with $CT_{min}$, so that ants at high elevations foraged at lower temperatures resulting in similar emergence times across elevations [18]. Similarly, *Temnothorax curvispinosus* ants in urban heat islands exhibited a higher $CT_{max}$ and $CT_{min}$ than their rural

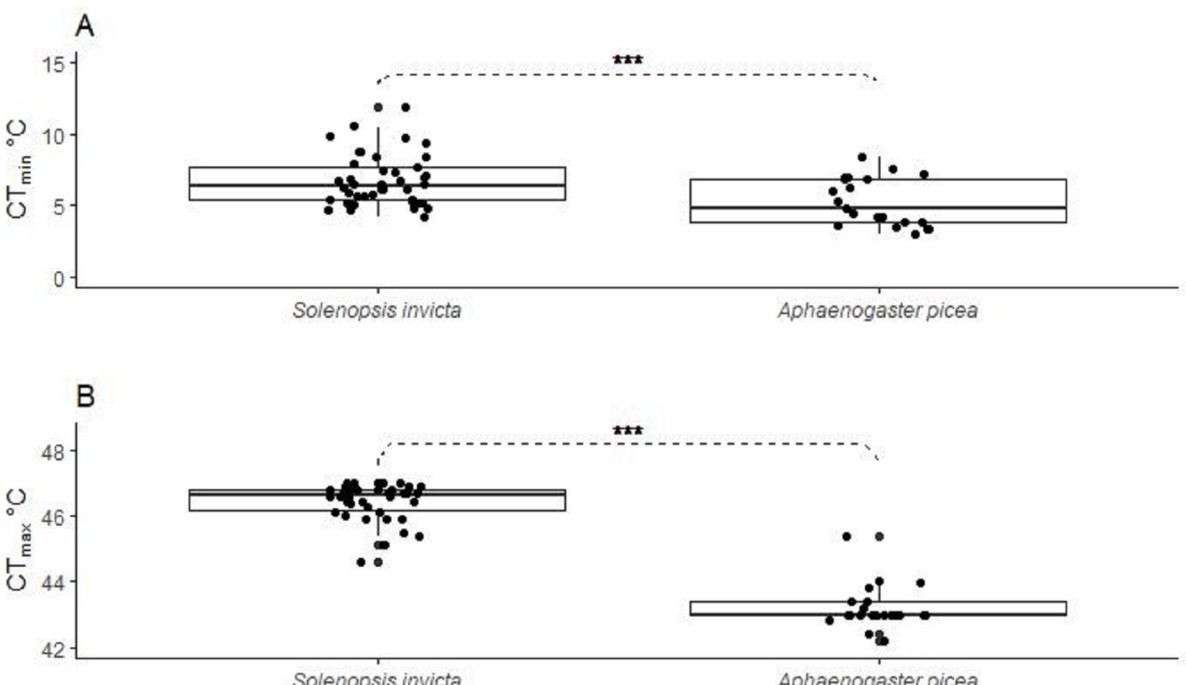

**Fig 2. Boxplots showing *Solenopsis invicta* and *Aphaenogaster picea* cold (CT$_{min}$; 2A) and heat (CT$_{max}$; 2B) tolerance.** Each boxplot includes the median (solid line) and upper and lower percentiles (25$^{th}$ and 75$^{th}$) with error bars indicating outliers. Asterisks indicate significance of *p-value* < 0.001.

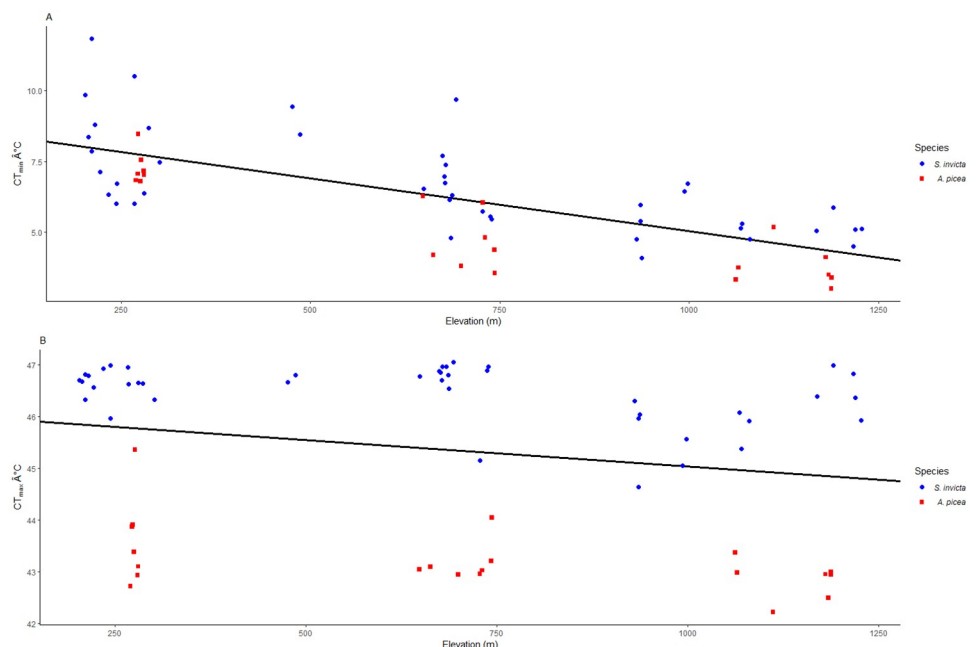

**Fig 3. Regression plots showing *Solenopsis invicta* and *Aphaenogaster picea* CT$_{min}$ with elevation (A), and CT$_{max}$ with elevation (B).**

counterparts in colder environments [19]. However, Warren et al. [21] found the opposite pattern in *Brachyponera chinensis* invasions in the same study region used here. *Brachyponera chinensis* is a non-native ant from Asia, and has invaded the forests of North and South Carolina and Georgia. Populations found at high elevations in the Southern Appalachian Mountains do not appear to persist, and Warren et al. [21] found that this pattern is explained by the ants' apparent inability to acclimate to colder temperatures (i.e., its cold tolerance does not change with elevation.

*Solenopsis invicta* exhibited higher heat tolerance than *A. picea*, and *A. picea* exhibited greater cold tolerance than *S. invicta*. These results were somewhat predictable given that *S. invicta* originates from a subtropical climate and *A. picea* is temperate. Moreover, the difference likely is exacerbated by the microhabitats that the two species occupy: *A. picea* is a woodland ant that favors shaded canopy understory environments, whereas *S. invicta* is a thermophilic species that thrives in highly disturbed environments with full sun [3, 14]. What is interesting however is that the heat and cold tolerances of both species decreased similarly with increased elevation, indicating that the non-native *S. invicta* ants had the same ability to adapt or acclimatize as the native *A. picea* ants.

We found the mean *S. invicta* $CT_{max}$ (46.4 ± 0.1˚C) to be similar to that found in other studies: 50.6 ˚C [22] and 46.5 ˚C (for large workers [23]). The mean $CT_{min}$ for *S. invicta* ants in our study (6.8 ± 0.1˚C) was higher than that found in low-elevation ants collected by Bentley et al. [22]: 4.1 ˚C. The difference may lie in research methodology. Moreover, we found no appreciable difference in thermal tolerance between the smallest (minors) and largest (majors) polymorphic workers, whereas the relationship between *S. invicta* size and thermal tolerance are mixed for other studies [22, 23]. This suggests that there may be regional differences in how body size of *S. invicta* affects thermal tolerance, or that differences in research methodologies may affect cross-study comparisons.

We found that a subtropical invasive ant can acclimate to colder temperatures than those found in its home range, and global change also may facilitate further spread of *S. invicta* to still greater latitudes and elevations as temperatures increase, as has occurred in other insects such as some Lepidoptera and Odonata [24, 25]. Warren and Chick [25] investigated the relationship between thermal tolerance limits and distribution shifts in two woodland ants, *A. picea* and *A. rudis*. The lower elevation *A. rudis* consistently exhibited temperature tolerance limits 2 ˚C higher than that of the higher elevation species, *A. picea*, in both minimum and maximum thermal tolerance testing. With warming, however, the lower elevation species displaced the higher species. Hence, as temperatures rise, more competitive ant species may be able to shift their distribution into novel habitats where they were once unable to persist and displace native and/or less competitive ants [4, 26].

*Solenopsis invicta* preferably invades anthropogenically disturbed habitats [27], and increasing fragmentation in the Blue Ridge Escarpment region [28] may have facilitated its invasion as much as thermal acclimation. As noted, we observed no *S. invicta* colonies in undisturbed forest, a haven for *A. picea*. Certainly, anthropogenic disturbance is not limited to the Blue Ridge Escarpment, and our results suggest that the ant's cold temperature tolerance will not limit its ability to utilize disturbed habitats northward of its current range. In conjunction with climate warming, a much greater portion of the eastern U.S. may be subject to *S. invicta* invasion.

## Supporting information

**S1 Fig. Regression plot of lipid content (g) of *Solenopsis invicta* ants collected along an elevational gradient.**
(TIF)

## Acknowledgments

We thank Highlands Biological Station, Coweeta Hydrologic Laboratory, and Western Carolina University for the use of laboratory space and supplies. We also thank Dr. Thomas Martin for his assistance with experimental design and statistical analysis, and Sonya Bayba, Kevin Krupp, and Kyle Pursel for their field assistance. The manuscript was improved by the helpful comments and criticisms of two anonymous reviewers.

## Author Contributions

**Conceptualization:** A. J. Lytle, J. T. Costa, R. J. Warren, II.

**Data curation:** A. J. Lytle.

**Formal analysis:** A. J. Lytle, R. J. Warren, II.

**Investigation:** A. J. Lytle.

**Methodology:** A. J. Lytle, R. J. Warren, II.

**Project administration:** A. J. Lytle, J. T. Costa.

**Resources:** J. T. Costa, R. J. Warren, II.

**Supervision:** J. T. Costa, R. J. Warren, II.

**Writing – original draft:** A. J. Lytle.

**Writing – review & editing:** A. J. Lytle, J. T. Costa, R. J. Warren, II.

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
