## [Decision Letter · Decision Letter 0]

14 Nov 2019

PONE-D-19-27277

Invasion and high-elevation acclimation of the red imported fire ant, Solenopsis invicta, in the southern Blue Ridge escarpment region

PLOS ONE

Dear Ms. Lafferty,

Thank you for submitting your manuscript to PLOS ONE. After careful consideration, we feel that it has merit but does not fully meet PLOS ONE’s publication criteria as it currently stands. Therefore, we invite you to submit a revised version of the manuscript that addresses the points raised during the review process.

Both reviewers have raised one critical issue. The first reviewer insists that data on the temperatures that colonies were experiencing must be included in the manuscript. The second reviewer highlights some issues with the third part of your study that also warrant some serious consideration.

We would appreciate receiving your revised manuscript by Dec 29 2019 11:59PM. To enhance the reproducibility of your results, we recommend that if applicable you deposit your laboratory protocols in protocols.io, where a protocol can be assigned its own identifier (DOI) such that it can be cited independently in the future. For instructions see: http://journals.plos.org/plosone/s/submission-guidelines#loc-laboratory-protocols

We look forward to receiving your revised manuscript.

Kind regards,

Olav Rueppell

Academic Editor

PLOS ONE

Journal Requirements:

1. In your Methods section, please provide additional location information of the collection sites, including geographic coordinates for the data set if available.

2. In your Methods section, please provide additional information regarding the permits you obtained for the work. Please ensure you have included the full name of the authority that approved the collection sites access and, if no permits were required, a brief statement explaining why.

3. Thank you for including the foloowing funding information within your acknowledgements section of your manuscript; "This research was supported in part by a Grant-in-Aid of Research and Ralph Sargent Scholarship provided by the Highlands Biological Foundation, Inc."

Additionally, because some of your funding information pertains to [commercial funding//patents], we ask you to provide an updated Competing Interests statement, declaring all sources of commercial funding.

In your Competing Interests statement, please confirm that your commercial funding does not alter your adherence to PLOS ONE Editorial policies and criteria by including the following statement: "This does not alter our adherence to PLOS ONE policies on sharing data and materials.” as detailed online in our guide for authors  http://journals.plos.org/plosone/s/competing-interests.  If this statement is not true and your adherence to PLOS policies on sharing data and materials is altered, please explain how.

Please include the updated Competing Interests Statement and Funding Statement in your cover letter. We will change the online submission form on your behalf.

4. We note that [Figure(s) 1] in your submission contain [map/satellite] images which may be copyrighted. All PLOS content is published under the Creative Commons Attribution License (CC BY 4.0), which means that the manuscript, images, and Supporting Information files will be freely available online, and any third party is permitted to access, download, copy, distribute, and use these materials in any way, even commercially, with proper attribution. For these reasons, we cannot publish previously copyrighted maps or satellite images created using proprietary data, such as Google software (Google Maps, Street View, and Earth). For more information, see our copyright guidelines: http://journals.plos.org/plosone/s/licenses-and-copyright.

1.    You may seek permission from the original copyright holder of Figure(s) [1] to publish the content specifically under the CC BY 4.0 license. 

Reviewers' comments:

Reviewer's Responses to Questions

**Comments to the Author**

1. Is the manuscript technically sound, and do the data support the conclusions?

Reviewer #1: Yes

Reviewer #2: Partly

2. Has the statistical analysis been performed appropriately and rigorously? 

Reviewer #1: Yes

Reviewer #2: Yes

3. Have the authors made all data underlying the findings in their manuscript fully available?

Reviewer #1: Yes

Reviewer #2: No

4. Is the manuscript presented in an intelligible fashion and written in standard English?

Reviewer #1: Yes

Reviewer #2: Yes

5. Review Comments to the Author

Reviewer #1: This is an important addition to our understanding of the ongoing invasion of the red imported fire ant around the world. The manuscript is generally well-written and the methods and results are sound and succinctly reported. The introduction and conclusion include testable hypotheses and discussion of results that are supported by the data presented.

My suggestions for change are as follows:

Why are there no data included either as a table or figure for the soil temperature profiles in the results? This is an important part of the study - what were the actual soil temperature profiles in open and thermal mass sites? Specifically, those data will be helpful for understanding weekly and monthly min and max temperatures experienced within the colonies (within the limitations of the 6 measurements per day). This is especially important as the conditions that the colony is experiencing (and is tolerant of) is ultimately much more important than what foraging workers can tolerate. Acclimation for a eusocial insect is much better measured for colonies and including those data will show what range of temperatures that colonies are experiencing. In my opinion, those data must be included for this to be publishable.

Lines 304-317. In a series of papers Tschinkel and King have provided a clear picture of how and likely why fire ant queens disperse to and select sites. Specifically, they are open habitat (particularly early successional habitats with no canopy cover) specialists. Queens actively select these sites for founding after mating. These references provide strong support for the ideas put forward in this section of the manuscript and also provide temperature ranges that colonies experience in Florida that might make for a useful comparison with the data that should be included in this manuscript. The references are:

Tschinkel, W.R. and J.R. King. 2013.The role of habitat in the persistence of fire ant populations. PLOS ONE. 8(10): e78580.

King, J.R. and W.R. Tschinkel. 2016. Experimental evidence that dispersal drives ant community assembly in human-altered ecosystems. Ecology 97: 236-249.

Tschinkel, W.R. and J.R. King. 2017. Abiotic and biotic factors limiting colony founding success for the fire ant, Solenopsis invicta. Functional Ecology 31: 955-964.

Reviewer #2: Invasion and high-elevation acclimation of the red imported fire ant, Solenopsis invicta,

in the southern Blue Ridge escarpment region

The goal of this study was to determine whether Solenopsis invicta (RIFA) colonies at high elevation have acclimated to high elevation environment. The author investigated three aspects of RIFA’s adaptation to high elevation environments (i) temperature tolerance of RIFA and native A. picea workers collected at different elevation, (ii) Measurement of colony lipid content at different elevation (iii) Measurement of soil temperature at high elevation of colonies near or away from a potential thermal mass.

It is rare to come across such a nicely written and concise paper. I really enjoyed reading it and I found the results quite interesting. I think section (i) of this work is the most interesting as the results are nice and clear. Section (ii) was a nice addition, although you did not find differences in elevation and lipid contents and did not report lipid contents results other than the non-significance. I’m not as enthusiastic for section (iii) and I think it would improve the quality of the paper to remove it or, if you decide to keep it, keep it short and rewrite the results and their significance. The main assumption for this part of your work is that proximity to thermal masses will increase nest temperature. Your results (L258-259) have shown that it is the other way around and that minimum soil temperatures were higher for colonies away from a radiant heat mass. You did not report any results about maximum and average soil temperatures. In any case, the original assumption is not validated so statements such as L47-48 do not make much sense as thermal masses were not, in fact, thermal masses. Reporting the results more accurately would be something like “Minimum soil temperatures were higher for colonies sited away from thermal masses”. Shade and exposition should have been taken into account in the analysis. It would have improved the strength of this work to add these variables to your model. However, you only selected nine paired colonies, a very low number of replicates to build such models. Additionally, it is unclear why you chose to only consider colonies at high elevation for this part of your work. You also cannot state in your results that colonies did not appear to select nesting sites near thermal masses. This is because you did not look at S. invicta queens nest selection. In fact, you cannot be sure that queens would have selected this nest for other reasons e.g. access to resources, predation risk, competition etc or if they selected the site at all. It would have been a nice addition to the paper to look at foraging time for S. invicta and A. picea at low and high elevation to determine whether colony behaviour changed with elevation.

Abstract

The abstract is nicely written and clear.

43-44: I find this sentence difficult to read. It should be rewritten to improve clarity.

Introduction

The introduction is short and easy to read. It introduces the paper well. I only have a few comments.

I think it would be interesting to mention the observed and potential impacts of S. invicta in the Blue Ridge escarpment region. Is the invasion threatening important ecosystems? Have you seen colonies penetrating undisturbed habitats?

L 59-63: Above you state that S. invicta cannot colonize areas with temperatures below -3.7°C but you do not give the minimal temperatures for the Blue Ridge escarpment region and the Southern Appalachian Mountains. I was also unclear about the difference between the two regions. Is the Blue Ridge escarpment part of the Southern Appalachian? It would be useful to clarify for those of us who are not familiar with this region.

L59: “In recent years” How long ago exactly?

L64-66 How did you confirm the persistence of colonies through the winter season at high elevation? I did not see any mention of observing colony persistence through the winter in the methods or results.

L72: delete “much”. Even though you give more details about both study species in the methods, I think a short sentence about A. picea’s current distribution would improve this paragraph. I think you should also state why you chose A. picea for this work rather than another native ant.

L76-77 change to “potential shorter optimal foraging time” (as you have not shown that the optimal foraging time of S. invicta is shorter at high elevation) and add references.

L80-94 Considering the limitation of your results, I think it would be best to drastically shorten this paragraph should you choose to leave this experiment in the paper. You should also explain why you only looked at minimum soil temperature.

L93-94 According to your methods, you did not in fact determine whether colonies at high elevation would be closer to thermal masses than at low elevation.

Methods

I found the methods confusing and unclear. I think it needs a major re-write.

Study species

I think some of the details given about S. invicta are not relevant to your study and could be shortened.

L101 Here and elsewhere. Avoid citing books as much as possible, it is generally better to cite journal articles that have been through a peer-review process.

L 101-104: You need to define disturbance here and give examples.

L109 and elsewhere: Personally, I would regard a scientific species name as singular as you refer here to one species and not several individuals. Both options are grammatically correct so it’s up to you.

L110: “many native understory herbs” Be more specific as to the number of species and give a couple of examples.

Collection sites

I think the clarity of the methods would be greatly improved if you added the site description and collection protocol with each experiment instead of having it in a separate section. The current structure of the methods is very difficult to follow, and I constantly had to switch back and forth between pages to understand the work that was done. The collection for the lipid analysis was not mentioned in this section.

L149 How big were the mounds and how similar in size were they to each other? I think you should give an indication of colony size for each experiment.

Figure 1: Add the meaning of the state abbreviations in the legend. You should also modify the map to show the distribution of low, middle and high elevation sites.

L127 and elsewhere It is unclear what you mean by locality. Do you mean the three elevational ranges? If so, “locality” does not seem appropriate.

L147 Why were the colonies for this work not the same as the thermal tolerance and lipid analyses colonies?

L151-157 This part of your protocol is confusing. I think you should add a map for this work or add the collection points for this work in your current map as the name of the sites are not relevant to the reader. I am confused by what you mean by “three main sites”. Here and elsewhere you use “site” to describe different things (e.g. your collection sites and the sites selected by a queen). You cannot use the same term to describe this very different things or the reader will be confused by your experimental design.

Thermal tolerance

I think this section could be shortened and better structured to improve clarity. It is a bit clunky in its current form. The number of major and minor workers tested needs to be mentioned whenever you mention testing S. invicta workers. It is lacking at L177 and L166.

L163-164: Give the total number of S. invicta major, minor and A. picea workers tested

L164 and 179: You repeat that you tested 15 A. picea workers L179. This should be earlier in this section (around L164) and the number of A. picea colonies collected should be stated from the start of this paragraph.

Lipid analysis

L184 Where were the ants collected from? How many ants did you collect in total?

L187: Specify in what way the protocol was modified.

Thermal effects of nest site selection

I have already commented on your nest site selection experiment in my general comments. I have a couple additional comments.

Here and in the results: You cannot call this part of your work “Thermal effects of nest site selection” as you haven’t shown that queens have selected nesting sites and that colony presence is not random.

L194-195 What do you mean by “detailed description”? What did you record?

L197 Were the two iButtons for each colony placed together in the plastic bags?

Results

Thermal tolerance

The results need to be more descriptive by specifying the temperature for each statement. For example, in the first paragraph, give us the average CTmax for minor and major workers. In the second paragraph, it would good to specify for example, “the thermal tolerance of ants found at higher elevation increased by xx°C compared to ants at lower elevation”. Don’t use as many decimals as in line 259, one or two is sufficient.

L234: “a larger overall range” Specify the range.

Figure 2: I think it would be more informative to have one regression line per species. The reader would be able to visually determine that both species respond the same way.

Figure 3: Add significance stars on the graphs and the statistical tests in the title. Figure 3A: The values for the y axis do not go high enough (i.e. maximum value is 10 but it should be somewhere around 12).

Lipid analysis

You need to add information about lipid content values along the gradient. A figure in the supplementary material would also be useful.

Thermal effects of nest site selection

I have already commented on this in the general comments. I have one additional major point: You have not reported any results for your observations described L203-205.

Discussion

The discussion is short and concise, and I generally enjoyed reading it apart for statements about thermal masses and nesting site selection which I will not comment on again. Another one of my concern with the discussion is the lack of references and examples in some paragraphs. I also found that some of your results could be furthered discussed to increase the impact of your paper.

The last paragraph should be more developed. Specifically, what do your results mean for studies looking at the future distribution of S. invicta under climate change scenario (e.g. Bertelsmeier, Cleo, et al. "Worldwide ant invasions under climate change." Biodiversity and conservation 24.1 (2015): 117-128).

Just a few more interrogations: How do your CTmin and CTmax values for S. invicta compare to values obtained for this species in previous studies? Have other invasive ants been found to have a shift in thermal tolerance at different elevation? The CTmin value you found for S. invicta was about 6C. What happens to the workers when the temperature is lower? Do they remain in the nest and is the temperature higher in the nest? Have you determined the lethal thermal maxima and lethal thermal minima.

L276-277 “other studies” There is only one reference for this paragraph. Give more references and examples.

L281-283 You need to make a better case with references to justify this hypothesis

L294-296 Add reference

L296-298 Give references and examples. “a wider range of (…) temperatures” Compared to what? The use of “survival tool” is odd. “survival tool” for what? The colony or workers? What about colony health and reproduction?

L299-301 This sentence is long and difficult to understand. It should be rewritten for clarity.

L319 Shouldn’t it be “as temperature warms up” (or even better “increases”) instead of “as temperature warms”?

L319 What other groups of animals? Give examples

L320-321 Give more details and examples

L323 “as the climate warms” I find this odd, can you rephrase it?

L323 “have serious implications for S. invicta”. Avoid using “serious” as it is unclear what you mean here. As it stands, it seems to mean that higher temperature will negatively affect S. invicta. A reference is also lacking.

L325 What other species? Give references

L334 “serious implications” same comment as above. Be more descriptive.

L335 “other ant species” Do you mean invasive ants?

L337-341 Can you detail how you came up with this change in range?

6. PLOS authors have the option to publish the peer review history of their article (what does this mean?). If published, this will include your full peer review and any attached files.

Reviewer #1: No

Reviewer #2: No

---

## [Author Response · Author response to Decision Letter 0]

19 Jan 2020

COMMENTS TO THE AUTHOR

Handling Editor Comments:

Thank you for submitting your manuscript to PLOS ONE. After careful consideration, we feel that it has merit but does not fully meet PLOS ONE’s publication criteria as it currently stands. Therefore, we invite you to submit a revised version of the manuscript that addresses the points raised during the review process.

Both reviewers have raised one critical issue. The first reviewer insists that data on the temperatures that colonies were experiencing must be included in the manuscript. The second reviewer highlights some issues with the third part of your study that also warrant some serious consideration.

1. In your Methods section, please provide additional location information of the collection sites, including geographic coordinates for the data set if available.

RESPONSE: We added geographic coordinates for the main collection sites in the Methods section. 

2. In your Methods section, please provide additional information regarding the permits you obtained for the work. Please ensure you have included the full name of the authority that approved the collection sites access and, if no permits were required, a brief statement explaining why.

RESPONSE: No permits were required for collecting. We had permission for collecting on private lands such as the Lonesome Valley Residential Community and on private lands that were in the high elevation areas. All other areas were in public places such as parking lots and on the side of the road. 

3. Thank you for including the following funding information within your acknowledgements section of your manuscript; "This research was supported in part by a Grant-in-Aid of Research and Ralph Sargent Scholarship provided by the Highlands Biological Foundation, Inc."

Additionally, because some of your funding information pertains to [commercial funding//patents], we ask you to provide an updated Competing Interests statement, declaring all sources of commercial funding.

In your Competing Interests statement, please confirm that your commercial funding does not alter your adherence to PLOS ONE Editorial policies and criteria by including the following statement: "This does not alter our adherence to PLOS ONE policies on sharing data and materials.” as detailed online in our guide for authors http://journals.plos.org/plosone/s/competing-interests. If this statement is not true and your adherence to PLOS policies on sharing data and materials is altered, please explain how.

Please include the updated Competing Interests Statement and Funding Statement in your cover letter. We will change the online submission form on your behalf.

RESPONSE: We have provided our amended funding statement below:

Amanda Lafferty (A.J.L) received funding from a Grant-in-Aid of Research and Ralph Sargent Scholarship provided by the Highlands Biological Foundation, Inc (Grant number HBS-GIA-2017-05). The funders had no role in study design, data collection and analysis, decision to publish, or preparation of the manuscript. This does not alter our adherence to PLOS ONE policies on sharing data and materials.

Highlands Biological Station URL: https://highlandsbiological.org/

4. We note that [Figure(s) 1] in your submission contain [map/satellite] images which may be copyrighted. All PLOS content is published under the Creative Commons Attribution License (CC BY 4.0), which means that the manuscript, images, and Supporting Information files will be freely available online, and any third party is permitted to access, download, copy, distribute, and use these materials in any way, even commercially, with proper attribution. For these reasons, we cannot publish previously copyrighted maps or satellite images created using proprietary data, such as Google software (Google Maps, Street View, and Earth). For more information, see our copyright guidelines: http://journals.plos.org/plosone/s/licenses-and-copyright.

You may seek permission from the original copyright holder of Figure(s) [1] to publish the content specifically under the CC BY 4.0 license. 

RESPONSE: All the data used in this map is public domain. We updated the figure caption with the following copyright permission statement: 

“Reprinted from National Boundaries Dataset, 3DEP Elevation Program (https://www.usgs.gov/core-science-systems/ngp/3dep) with permission from USGS, public domain 2017, as well as from the 2017 TIGER/Line Shapefiles (https://www.census.gov/geo/maps-data/data/tiger-line.html) with permission from the U.S. Census Bureau, public domain 2017." 

Comments to the Author

Reviewer #1: This is an important addition to our understanding of the ongoing invasion of the red imported fire ant around the world. The manuscript is generally well-written and the methods and results are sound and succinctly reported. The introduction and conclusion include testable hypotheses and discussion of results that are supported by the data presented.

My suggestions for change are as follows:

Why are there no data included either as a table or figure for the soil temperature profiles in the results? This is an important part of the study - what were the actual soil temperature profiles in open and thermal mass sites? Specifically, those data will be helpful for understanding weekly and monthly min and max temperatures experienced within the colonies (within the limitations of the 6 measurements per day). This is especially important as the conditions that the colony is experiencing (and is tolerant of) is ultimately much more important than what foraging workers can tolerate. Acclimation for a eusocial insect is much better measured for colonies and including those data will show what range of temperatures that colonies are experiencing. In my opinion, those data must be included for this to be publishable.

Lines 304-317. In a series of papers Tschinkel and King have provided a clear picture of how and likely why fire ant queens disperse to and select sites. Specifically, they are open habitat (particularly early successional habitats with no canopy cover) specialists. Queens actively select these sites for founding after mating. These references provide strong support for the ideas put forward in this section of the manuscript and also provide temperature ranges that colonies experience in Florida that might make for a useful comparison with the data that should be included in this manuscript. The references are:

Tschinkel, W.R. and J.R. King. 2013.The role of habitat in the persistence of fire ant populations. PLOS ONE. 8(10): e78580.

King, J.R. and W.R. Tschinkel. 2016. Experimental evidence that dispersal drives ant community assembly in human-altered ecosystems. Ecology 97: 236-249.

Tschinkel, W.R. and J.R. King. 2017. Abiotic and biotic factors limiting colony founding success for the fire ant, Solenopsis invicta. Functional Ecology 31: 955-964.

RESPONSE: We decided to remove the thermal mass analysis along with other sections describing part III, as suggested by Reviewer 2, however, we have included soil temperature profile data in our Dryad Digital Repository along with the rest of the data from this study. 

Reviewer #2: Invasion and high-elevation acclimation of the red imported fire ant, Solenopsis invicta, in the southern Blue Ridge escarpment region

The goal of this study was to determine whether Solenopsis invicta (RIFA) colonies at high elevation have acclimated to high elevation environment. The author investigated three aspects of RIFA’s adaptation to high elevation environments (i) temperature tolerance of RIFA and native A. picea workers collected at different elevation, (ii) Measurement of colony lipid content at different elevation (iii) Measurement of soil temperature at high elevation of colonies near or away from a potential thermal mass.

It is rare to come across such a nicely written and concise paper. I really enjoyed reading it and I found the results quite interesting. I think section (i) of this work is the most interesting as the results are nice and clear. Section (ii) was a nice addition, although you did not find differences in elevation and lipid contents and did not report lipid contents results other than the non-significance. I’m not as enthusiastic for section (iii) and I think it would improve the quality of the paper to remove it or, if you decide to keep it, keep it short and rewrite the results and their significance. The main assumption for this part of your work is that proximity to thermal masses will increase nest temperature. Your results (L258-259) have shown that it is the other way around and that minimum soil temperatures were higher for colonies away from a radiant heat mass. You did not report any results about maximum and average soil temperatures. In any case, the original assumption is not validated so statements such as L47-48 do not make much sense as thermal masses were not, in fact, thermal masses. Reporting the results more accurately would be something like “Minimum soil temperatures were higher for colonies sited away from thermal masses”. Shade and exposition should have been taken into account in the analysis. It would have improved the strength of this work to add these variables to your model. However, you only selected nine paired colonies, a very low number of replicates to build such models. Additionally, it is unclear why you chose to only consider colonies at high elevation for this part of your work. You also cannot state in your results that colonies did not appear to select nesting sites near thermal masses. This is because you did not look at S. invicta queens nest selection. In fact, you cannot be sure that queens would have selected this nest for other reasons e.g. access to resources, predation risk, competition etc or if they selected the site at all. It would have been a nice addition to the paper to look at foraging time for S. invicta and A. picea at low and high elevation to determine whether colony behaviour changed with elevation.

Abstract

The abstract is nicely written and clear.

43-44: I find this sentence difficult to read. It should be rewritten to improve clarity.

RESPONSE: We changed the sentence to read “Both S. invicta and A. picea similarly exhibited lower thermal tolerances for colder temperatures when moving up the elevational gradient, with A. picea consistently exhibiting a lower thermal tolerance overall.”

Introduction

The introduction is short and easy to read. It introduces the paper well. I only have a few comments.

I think it would be interesting to mention the observed and potential impacts of S. invicta in the Blue Ridge escarpment region. Is the invasion threatening important ecosystems? Have you seen colonies penetrating undisturbed habitats?

RESPONSE: Our research was focused on regional-scale patterns in S. invicta invasion, and hence we must be careful to not overrun our data. Moreover, we have not observed S. invicta in undisturbed habitats. However, we added the following to the Discussion: “Solenopsis invicta invades anthropogenically disturbed habitats (King and Tschinkel 2008), and increased fragmentation in the Blue Ridge Escarpment region (Turner et al. 2003) suggest that the region may experience still greater impacts of the non-native ant. Moreover, at high elevations, the open features of sensitive rock outcrop habitats may make them vulnerable to S. invicta invasions.”

L 59-63: Above you state that S. invicta cannot colonize areas with temperatures below -3.7°C but you do not give the minimal temperatures for the Blue Ridge escarpment region and the Southern Appalachian Mountains. I was also unclear about the difference between the two regions. Is the Blue Ridge escarpment part of the Southern Appalachian? It would be useful to clarify for those of us who are not familiar with this region.

RESPONSE: To give examples of temperatures of the Blue Ridge escarpment region, we added an example from NOAA: “Within the last five years at least, S. invicta colonies have been observed at elevations > 1220 m in the Blue Ridge Escarpment where average low temperatures are < 4.5 °C (U.S. Climate Data 2019; Yeary-Johnson 2014), and in fact, the average temperature for Macon county (where part of this study was conducted) in January 2019 was 2.9 °C (NOAA 2019).”

The Blue Ridge Escarpment is a subset of the Southern Appalachian mountains but we changed the any instances of “Southern Appalachian mountains” to the more specific “Blue Ridge Escarpment” to clarify for readers who are not familiar with the area. We also explained what this term means in the introduction: “As such, the southern Blue Ridge Escarpment region in western North Carolina, U. S. (the zone of abrupt change in elevation between the Blue Ridge and Piedmont physiographic provinces with a vertical relief of 400 m to ca. 760 m)…”

L59: “In recent years” How long ago exactly?

RESPONSE: We updated this sentence to read “Within the last five years at least…”

L64-66 How did you confirm the persistence of colonies through the winter season at high elevation? I did not see any mention of observing colony persistence through the winter in the methods or results.

RESPONSE: This was removed along with other sections describing part III as suggested by Reviewer 2. 

L72: delete “much”. Even though you give more details about both study species in the methods, I think a short sentence about A. picea’s current distribution would improve this paragraph. I think you should also state why you chose A. picea for this work rather than another native ant.

RESPONSE: We deleted “much” and included the suggested information in the Introduction: “Aphaenogaster picea are native to the deciduous forests in the eastern U. S. and are not only the most abundant ants in this region, but are important ecologically as seed dispersers of many native understory herbs such as Sanguinaria, Trillium, and Hepatica spp. (Ness et al. 2009; Lubertazzi 2012).”

L76-77 change to “potential shorter optimal foraging time” (as you have not shown that the optimal foraging time of S. invicta is shorter at high elevation) and add references.

RESPONSE: This was changed as suggested. 

L80-94 Considering the limitation of your results, I think it would be best to drastically shorten this paragraph should you choose to leave this experiment in the paper. You should also explain why you only looked at minimum soil temperature.

RESPONSE: This was removed along with other sections describing part III as suggested by Reviewer 2. 

L93-94 According to your methods, you did not in fact determine whether colonies at high elevation would be closer to thermal masses than at low elevation.

RESPONSE: This was removed along with other sections describing part III as suggested by Reviewer 2. 

Methods

I found the methods confusing and unclear. I think it needs a major re-write.

Study species

I think some of the details given about S. invicta are not relevant to your study and could be shortened.

RESPONSE: We took out lines 104-108 to shorten the section on S. invicta as a study species.

L101 Here and elsewhere. Avoid citing books as much as possible, it is generally better to cite journal articles that have been through a peer-review process.

RESPONSE: We mostly cited “Fire Ants” by Tschinkel, which we regard as a cornerstone of S. invicta research. However, we agree that primary research is most important, and we tried to include primary research with the book citations.

L 101-104: You need to define disturbance here and give examples.

RESPONSE: We added examples and changed the sentence to:

“Habitat disturbance such as the construction of buildings and roads or deforestation is crucial for the establishment of many invasive species…” 

L109 and elsewhere: Personally, I would regard a scientific species name as singular as you refer here to one species and not several individuals. Both options are grammatically correct so it’s up to you.

RESPONSE: We prefer to keep the wording as is. 

L110: “many native understory herbs” Be more specific as to the number of species and give a couple of examples.

RESPONSE: We added examples of understory herbs to this section: “Aphaenogaster picea are native to the deciduous forests in the eastern U. S. and are not only the most abundant ants in this region, but are important ecologically as seed dispersers of many native understory herbs such as Sanguinaria, Trillium, and Hepatica spp. (Ness et al. 2009; Lubertazzi 2012).” 

Collection sites

I think the clarity of the methods would be greatly improved if you added the site description and collection protocol with each experiment instead of having it in a separate section. The current structure of the methods is very difficult to follow, and I constantly had to switch back and forth between pages to understand the work that was done. The collection for the lipid analysis was not mentioned in this section.

RESPONSE: We clarified in the ‘collection sites’ section that the same ants used for the lipid analysis assays were from the same colonies and collected at the same time as the ants from the thermal tolerance assays. We prefer to keep the rest of the Methods as is. 

L149 How big were the mounds and how similar in size were they to each other? I think you should give an indication of colony size for each experiment.

RESPONSE: We collected subsamples of colonies and not full colonies since it is difficult to collect every ant in a S. invicta colony, and therefore difficult to gauge how big a colony is from the surface. 

Figure 1: Add the meaning of the state abbreviations in the legend. You should also modify the map to show the distribution of low, middle and high elevation sites.

RESPONSE: We added state abbreviations to the legend and changed the shape of the points on the map to reflect elevational distribution. 

L127 and elsewhere It is unclear what you mean by locality. Do you mean the three elevational ranges? If so, “locality” does not seem appropriate.

RESPONSE: We changed ‘locality’ to ‘elevational range’ as suggested. 

L147 Why were the colonies for this work not the same as the thermal tolerance and lipid analyses colonies?

RESPONSE: This was removed along with other sections describing part III as suggested by Reviewer 2. 

L151-157 This part of your protocol is confusing. I think you should add a map for this work or add the collection points for this work in your current map as the name of the sites are not relevant to the reader. I am confused by what you mean by “three main sites”. Here and elsewhere you use “site” to describe different things (e.g. your collection sites and the sites selected by a queen). You cannot use the same term to describe this very different things or the reader will be confused by your experimental design.

RESPONSE: This was removed along with other sections describing part III as suggested by the reviewer. 

Thermal tolerance

I think this section could be shortened and better structured to improve clarity. It is a bit clunky in its current form. The number of major and minor workers tested needs to be mentioned whenever you mention testing S. invicta workers. It is lacking at L177 and L166.

RESPONSE: We added the number of worker ants at every instance that is mentioned, and we shortened several sentences to improve clarity. 

L163-164: Give the total number of S. invicta major, minor and A. picea workers tested

RESPONSE: We added the total number of workers for each category as suggested. 

L164 and 179: You repeat that you tested 15 A. picea workers L179. This should be earlier in this section (around L164) and the number of A. picea colonies collected should be stated from the start of this paragraph.

RESPONSE: We added the number of colonies and the individuals collected for A. picea at the beginning of the section: “We tested 30 ants from each of the 42 S. invicta colonies and 15 ants form each of the 21 A. picea colonies. Solenopsis invicta are polymorphic, so we included the largest (major) and smallest (minor) workers of S. invicta colonies in our assay. We tested equal numbers of S. invicta majors (n=15) and minors (n=15), whereas A. picea only has one worker size, so we tested 15 total workers from each colony.”

Lipid analysis

L184 Where were the ants collected from? How many ants did you collect in total?

RESPONSE: We added that the ants were from the same colonies that we sampled for the thermal tolerance tests:

“Following thermal tolerance assays, the rest of the collected S. invicta ants were freeze-killed in a -80 °C freezer to use in the lipid analysis tests”. 

We also stated in the Methods section:

“We collected 300 worker ants from fourteen S. invicta colonies at each of three localities…” and “Two hundred ants from each colony were haphazardly selected from the samples and dried at 60 °C for 48 hours”.

L187: Specify in what way the protocol was modified.

RESPONSE: We changed the sentence to read:

 “The ants were weighed and dry mass recorded. Lipids were removed using a Soxhlet extractor following the of Smith and Tschinkel (2009) except that we sampled whole-colony lipids rather than individual ants.” 

Thermal effects of nest site selection

I have already commented on your nest site selection experiment in my general comments. I have a couple additional comments.

Here and in the results: You cannot call this part of your work “Thermal effects of nest site selection” as you haven’t shown that queens have selected nesting sites and that colony presence is not random.

RESPONSE: This was removed along with other sections describing part III as suggested by Reviewer 2. 

L194-195 What do you mean by “detailed description”? What did you record?

RESPONSE: This was removed along with other sections describing part III as suggested by Reviewer 2. 

L197 Were the two iButtons for each colony placed together in the plastic bags?

RESPONSE: This was removed along with other sections describing part III as suggested by Reviewer 2. 

Results

Thermal tolerance

The results need to be more descriptive by specifying the temperature for each statement. For example, in the first paragraph, give us the average CTmax for minor and major workers. In the second paragraph, it would good to specify for example, “the thermal tolerance of ants found at higher elevation increased by xx°C compared to ants at lower elevation”. Don’t use as many decimals as in line 259, one or two is sufficient.

RESPONSE: We added average CTmax for major and minor workers as suggested with the following statement: “The average CTmax was 46.6 °C for major workers and 46.3 °C for minor workers and the average CTmin was 6.9 °C for major workers and 6.6 °C for minor workers. Therefore, we did not separate the two size classes of worker ants in further analyses.”

We also added an example of the Ctmin difference: “On average the CTmin decreased by 2.8 °C for S. invicta and 3.5 °C for A. picea as elevations increased.”

We respectfully disagree on the number of decimals in reporting statistical results, and we prefer to report the standard three decimal places.

L234: “a larger overall range” Specify the range.

RESPONSE: We specified the overall range by adding the following details to this sentence: “…with S. invicta exhibiting a larger overall range (39.7 °C) than A. picea (37.9 °C) (SE = 0.352, t-value = -5.284, p-value < 0.001; Fig. 2C).”

Figure 2: I think it would be more informative to have one regression line per species. The reader would be able to visually determine that both species respond the same way.

RESPONSE: We understand the reviewer’s suggestion; however, we would like to keep the single regression line because it not only illustrates the general trends across elevation, but it provides a nice general threshold above and below which we see a distinct difference in species.

Figure 3: Add significance stars on the graphs and the statistical tests in the title. Figure 3A: The values for the y axis do not go high enough (i.e. maximum value is 10 but it should be somewhere around 12).

RESPONSE: We fixed figure 3A by increasing the Y axis to 15 and added significance stars on the graph and on the figure caption. 

Lipid analysis

You need to add information about lipid content values along the gradient. A figure in the supplementary material would also be useful.

RESPONSE: We have included a figure about lipid content in the supplementary material. 

Thermal effects of nest site selection

I have already commented on this in the general comments. I have one additional major point: You have not reported any results for your observations described L203-205.

RESPONSE: This was removed along with other sections describing part III as suggested by Reviewer 2. 

Discussion

The discussion is short and concise, and I generally enjoyed reading it apart for statements about thermal masses and nesting site selection which I will not comment on again. Another one of my concern with the discussion is the lack of references and examples in some paragraphs. I also found that some of your results could be furthered discussed to increase the impact of your paper.

The last paragraph should be more developed. Specifically, what do your results mean for studies looking at the future distribution of S. invicta under climate change scenario (e.g. Bertelsmeier, Cleo, et al. "Worldwide ant invasions under climate change." Biodiversity and conservation 24.1 (2015): 117-128).

RESPONSE: We did not suggest future distributions under climate change but indicated that our results show that S. invicta can spread to and occupy lower temperature habitats than currently understood and, of course, those lower temperatures may decrease with climate change.

Just a few more interrogations: How do your CTmin and CTmax values for S. invicta compare to values obtained for this species in previous studies? Have other invasive ants been found to have a shift in thermal tolerance at different elevation? The CTmin value you found for S. invicta was about 6C. What happens to the workers when the temperature is lower? Do they remain in the nest and is the temperature higher in the nest? Have you determined the lethal thermal maxima and lethal thermal minima.

RESPONSE: Cokendolpher and Phillips Jr. (1990) found that S. invicta collected in Texas had a Ctmax of 40.7 C, however the heating method was different than ours, so we did not include comparisons. 

We did not determine the lethal thermal maxima and lethal thermal minima.

L276-277 “other studies” There is only one reference for this paragraph. Give more references and examples.

RESPONSE: We added an example from Diamond SE, Chick L, Perez A, Strickler SA, Martin RA. Rapid evolution of ant thermal tolerance across and urban-rural temperature cline. Biological Journal of the Linnean Society. 2017;121: 248–257.

L281-283 You need to make a better case with references to justify this hypothesis

RESPONSE: We removed L281-283 from the manuscript. 

L294-296 Add reference

RESPONSE: We added references to this sentence: “A. picea is a woodland ant that favors shaded environments, whereas S. invicta is a thermophilic species that thrives in highly disturbed environments with full sun (Tschinkel 2006; Lubertazzi 2012).”

L296-298 Give references and examples. “a wider range of (…) temperatures” Compared to what? The use of “survival tool” is odd. “survival tool” for what? The colony or workers? What about colony health and reproduction?

RESPONSE: We removed these lines to reduce confusion. 

L299-301 This sentence is long and difficult to understand. It should be rewritten for clarity.

RESPONSE: We re-wrote the sentence to read: 

“There was a shift in CTmax on an elevational gradient as well. Solenopsis invicta had an overall higher heat tolerance than did A. picea as predicted, since S. invicta originates from a subtropical climate, but both species of ants exhibited downward shift of CTmax as elevations increased.”

L319 Shouldn’t it be “as temperature warms up” (or even better “increases”) instead of “as temperature warms”?

RESPONSE: We changed the wording to “as temperatures increase” as suggested. 

L319 What other groups of animals? Give examples

RESPONSE: We added examples from the study that we cited there: 

Sunday JM, Bates AE, Duly NK. Thermal tolerance and the global redistribution of animals. Nature Climate Change. 2012;2(9): 686–690.

L320-321 Give more details and examples

RESPONSE: We changed the sentence to read:

“Warming temperatures allow some species to expand their ranges at different rates and competitive species such as Aphaenogaster rudis and S. invicta often have a strong dispersal ability (Calcaterra et al. 2008; Urban et al. 2012; Warren et al. 2016).”

L323 “as the climate warms” I find this odd, can you rephrase it?

RESPONSE: We reworded parts of the discussion and removed this sentence to improve the flow. 

L323 “have serious implications for S. invicta”. Avoid using “serious” as it is unclear what you mean here. As it stands, it seems to mean that higher temperature will negatively affect S. invicta. A reference is also lacking.

RESPONSE: We took out the word ‘serious’ and changed the sentence to read:

“...and this could have implications for S. invicta, as they seem to be partially limited by minimum temperatures as shown in this study.”

L325 What other species? Give references

RESPONSE: We included the example of Aphaenogaster rudis with a reference. 

L334 “serious implications” same comment as above. Be more descriptive.

RESPONSE: We reworded parts of the discussion and removed this sentence to improve the flow.

L335 “other ant species” Do you mean invasive ants?

RESPONSE: We mean potentially all ant species.

L337-341 Can you detail how you came up with this change in range?

RESPONSE: We changed the final paragraph to “Solenopsis invicta invades anthropogenically disturbed habitats (King and Tschinkel 2008), and increased fragmentation in the Blue Ridge Escarpment region (Turner et al. 2003) suggest that the region may experience still greater impacts of the non-native ant. Moreover, at high elevations, the open features of sensitive rock outcrop habitats may make them vulnerable to S. invicta invasions. Combined with their apparent ability to acclimate or adapt to cooler temperatures and higher minimum temperatures predicted with global climate change, our results suggest that a much greater portion of the eastern U.S. may be subject to S. invicta invasion than is currently occupied.”

---

## [Decision Letter · Decision Letter 1]

11 Feb 2020

PONE-D-19-27277R1

Invasion and high-elevation acclimation of the red imported fire ant, Solenopsis invicta, in the southern Blue Ridge escarpment region of North America

PLOS ONE

Dear Ms. Lafferty,

Thank you for submitting your manuscript to PLOS ONE. After careful consideration, we feel that it has merit but does not fully meet PLOS ONE’s publication criteria as it currently stands. Therefore, we invite you to submit a revised version of the manuscript that addresses the points raised during the review process.

While the first reviewer is content with the first revision without further suggestion, the second reviewer has done an excellent job in thoroughly reviewing the revision and highlights a number of additional issues that I would like you to address.

We would appreciate receiving your revised manuscript by Mar 27 2020 11:59PM. To enhance the reproducibility of your results, we recommend that if applicable you deposit your laboratory protocols in protocols.io, where a protocol can be assigned its own identifier (DOI) such that it can be cited independently in the future. For instructions see: http://journals.plos.org/plosone/s/submission-guidelines#loc-laboratory-protocols

We look forward to receiving your revised manuscript.

Kind regards,

Olav Rueppell

Academic Editor

PLOS ONE

Reviewers' comments:

Reviewer's Responses to Questions

**Comments to the Author**

1. If the authors have adequately addressed your comments raised in a previous round of review and you feel that this manuscript is now acceptable for publication, you may indicate that here to bypass the “Comments to the Author” section, enter your conflict of interest statement in the “Confidential to Editor” section, and submit your "Accept" recommendation.

Reviewer #1: All comments have been addressed

Reviewer #2: (No Response)

2. Is the manuscript technically sound, and do the data support the conclusions?

Reviewer #1: (No Response)

Reviewer #2: Yes

3. Has the statistical analysis been performed appropriately and rigorously? 

Reviewer #1: (No Response)

Reviewer #2: Yes

4. Have the authors made all data underlying the findings in their manuscript fully available?

Reviewer #1: (No Response)

Reviewer #2: Yes

5. Is the manuscript presented in an intelligible fashion and written in standard English?

Reviewer #1: (No Response)

Reviewer #2: Yes

6. Review Comments to the Author

Reviewer #1: (No Response)

Reviewer #2: Invasion and high-elevation acclimation of the red imported fire ant, Solenopsis invicta,

in the southern Blue Ridge escarpment region of North America

The authors have responded to most of my comments to my satisfaction. A few of them still need attention and I have also made some additional comments. My main concern is that the discussion lacks references regarding previous work done on the thermal tolerance of S. invicta and other invasives. As a result, I find that the significance of this study has not been appropriately discussed.

Abstract

L45 “winter dormancy” I don’t think you have mentioned winter dormancy anywhere else. Edit this sentence or explain how this term is relevant to RIFA colonies in this region in the introduction and abstract.

Introduction

I think it would be interesting to mention the observed and potential impacts of S.

invicta in the Blue Ridge escarpment region. Is the invasion threatening important

ecosystems? Have you seen colonies penetrating undisturbed habitats?

RESPONSE: Our research was focused on regional-scale patterns in S. invicta

invasion, and hence we must be careful to not overrun our data. Moreover, we have

not observed S. invicta in undisturbed habitats. However, we added the following to the

Discussion: “Solenopsis invicta invades anthropogenically disturbed habitats (King and

Tschinkel 2008), and increased fragmentation in the Blue Ridge Escarpment region

(Turner et al. 2003) suggest that the region may experience still greater impacts of the

non-native ant. Moreover, at high elevations, the open features of sensitive rock

outcrop habitats may make them vulnerable to S. invicta invasions.”

The last two paragraphs of the discussion leave the reader with many questions instead of ending the paper nicely. See also my further comments on the discussion.

“Solenopsis invicta invades anthropogenically disturbed habitats (King and Tschinkel 2008) (…)” Add “preferably” as S. invicta has also been found to penetrate relatively undisturbed habitats.

“greater impacts” You need to describe what the current impacts are (if there are any) before mentioning “greater” impacts. I understand that you do not want to oversell your data, but is there any published evidence of RIFA impacts in the region or in similar ecosystems? If there is currently no evidence for impacts in the region, you should only mention that RIFA’s distribution may increase.

“sensitive rock outcrop habitats” I fail to see the relevance here as I do not think you have mentioned these habitats elsewhere in the paper (e.g. in the introduction). Are they undisturbed habitats? If so, and considering S. invicta preferably invades undisturbed habitats, is it likely this habitat is at risk of invasion? What makes these habitats sensitive? What organisms may be affected by a potential invasion? Have you seen S. invicta in these habitats? Perhaps it may be worth describing these habitats in the introduction where you describe study location.

The writing of these two paragraphs would also be greatly improved if it was more concise. e.g. “such as S. invicta shown here” is unnecessary

The fact that you have not observed invicta penetrating undisturbed environments at your sites is interesting for researchers working on the effect of disturbance on the habitat suitability for invasive species. You may want to add this observation somewhere as a personal observation.

L 59-63: Above you state that S. invicta cannot colonize areas with temperatures below

-3.7°C but you do not give the minimal temperatures for the Blue Ridge escarpment

region and the Southern Appalachian Mountains. I was also unclear about the

difference between the two regions. Is the Blue Ridge escarpment part of the Southern

Appalachian? It would be useful to clarify for those of us who are not familiar with this

region.

RESPONSE: To give examples of temperatures of the Blue Ridge escarpment region,

we added an example from NOAA: “Within the last five years at least, S. invicta

colonies have been observed at elevations > 1220 m in the Blue Ridge Escarpment

where average low temperatures are < 4.5 °C (U.S. Climate Data 2019; Yeary-

Johnson 2014), and in fact, the average temperature for Macon county (where part of

this study was conducted) in January 2019 was 2.9 °C (NOAA 2019).”

The Blue Ridge Escarpment is a subset of the Southern Appalachian mountains but

we changed the any instances of “Southern Appalachian mountains” to the more

specific “Blue Ridge Escarpment” to clarify for readers who are not familiar with the

area. We also explained what this term means in the introduction: “As such, the

southern Blue Ridge Escarpment region in western North Carolina, U. S. (the zone of

abrupt change in elevation between the Blue Ridge and Piedmont physiographic

provinces with a vertical relief of 400 m to ca. 760 m)…”

Information about the minimal temperature for the region is still missing. You state that RIFA cannot invade areas for which temperatures go below -3.7°C and then that “As such, the southern Blue Ridge Escarpment region (…) was projected as unsuitable habitat for S. invicta due to the cold temperature extremes at high elevations (Korzukhin 2001).” You need to tell the reader what these cold temperatures extremes are and the elevation at which they were recorded to make this information (i.e. that RIFA cannot invade regions with min temp. below -3.7C) relevant to your study.

L72: delete “much”. Even though you give more details about both study species in the

methods, I think a short sentence about A. picea’s current distribution would improve

this paragraph. I think you should also state why you chose A. picea for this work

rather than another native ant.

RESPONSE: We deleted “much” and included the suggested information in the

Introduction: “Aphaenogaster picea are native to the deciduous forests in the eastern

U. S. and are not only the most abundant ants in this region, but are important

ecologically as seed dispersers of many native understory herbs such as Sanguinaria,

Trillium, and Hepatica spp. (Ness et al. 2009; Lubertazzi 2012).”

Shouldn’t it be “one of the most abundant ants”?

L76-77 change to “potential shorter optimal foraging time” (as you have not shown that

the optimal foraging time of S. invicta is shorter at high elevation) (…).

RESPONSE: This was changed as suggested.

This was not changed.

Methods

L 101-104: You need to define disturbance here and give examples.

RESPONSE: We added examples and changed the sentence to:

“Habitat disturbance such as the construction of buildings and roads or deforestation is

crucial for the establishment of many invasive species…”

An ecological definition of disturbance with more specific examples and references is still needed. It is unclear whether you are defining disturbance broadly (e.g. including storms, floods) or restrict your definition to anthropogenic disturbance.

L149 How big were the mounds and how similar in size were they to each other? I

think you should give an indication of colony size for each experiment.

RESPONSE: We collected subsamples of colonies and not full colonies since it is

difficult to collect every ant in a S. invicta colony, and therefore difficult to gauge how

big a colony is from the surface.

If you are unable to give an estimation of colony size, specify whether you only collected workers from mature colonies or whether you also included incipient colonies.

L168 The ants were weighed and dry mass recorded. Lipids were removed using a Soxhlet extractor following the of Smith and Tschinkel (2009) except that we sampled whole-colony lipids rather than individual ants.

Change to “and their dry mass recorded”. A word is missing “Following the xxx of Smith (…)”

Figure 1: Add the full species name on the figure and delete this statement from the legend: “Solenopsis invicta colonies are marked in black and Aphaenogaster picea colonies are marked in white.”

Results

Thermal tolerance

The results need to be more descriptive by specifying the temperature for each

statement. For example, in the first paragraph, give us the average CTmax for minor

and major workers. In the second paragraph, it would good to specify for example, “the

thermal tolerance of ants found at higher elevation increased by xx°C compared to

ants at lower elevation”. Don’t use as many decimals as in line 259, one or two is

sufficient.

RESPONSE: We added average CTmax for major and minor workers as suggested

with the following statement: “The average CTmax was 46.6 °C for major workers and

46.3 °C for minor workers and the average CTmin was 6.9 °C for major workers and

6.6 °C for minor workers. Therefore, we did not separate the two size classes of worker

ants in further analyses.”

We also added an example of the Ctmin difference: “On average the CTmin decreased

by 2.8 °C for S. invicta and 3.5 °C for A. picea as elevations increased.”

We respectfully disagree on the number of decimals in reporting statistical results, and

we prefer to report the standard three decimal places.

Here and elsewhere throughout the paper, you need to add a measure of variation (e.g. standard deviation) with each average (including in the introduction L59-63). I agree on the number of decimals to report for statistical results, my comment about the number of decimals was meant for the temperatures you reported L259 of the original manuscript.

L198 “The CTmax for both species also decreased with elevation (SE < 0.001, t-value = -3.627, p-value < 0.001; Fig. 2B).”

Specify by how much CTmax decreased just as you did for CTmin.

L210 “Aphaenogaster picea had a lower CTmin than S. invicta (SE = 0.318, t-value = -4.150, p-value <0.001; Fig. 3A) whereas S. invicta had a higher CTmax than A. picea (SE = 0.15, t-value = -21.19,p-value < 0.001; Fig. 3B).”

Same comment as in my original review, give values for CTmin and CTmax for each species.

Lipid analysis

You need to add information about lipid content values along the gradient. A figure in

the supplementary material would also be useful.

RESPONSE: We have included a figure about lipid content in the supplementary

material.

You have not given lipid content values at different elevation in the main text. These values are important information regardless of your statistical results. You also did not reference your supplementary figure in the text.

Discussion

Just a few more interrogations: How do your CTmin and CTmax values for S. invicta

compare to values obtained for this species in previous studies? Have other invasive

ants been found to have a shift in thermal tolerance at different elevation? The CTmin

value you found for S. invicta was about 6C. What happens to the workers when the

temperature is lower? Do they remain in the nest and is the temperature higher in the

nest? Have you determined the lethal thermal maxima and lethal thermal minima.

RESPONSE: Cokendolpher and Phillips Jr. (1990) found that S. invicta collected in

Texas had a Ctmax of 40.7 C, however the heating method was different than ours, so

we did not include comparisons.

We did not determine the lethal thermal maxima and lethal thermal minima.

The significance of your results has not been fully explained in your discussion. That the methodology of Cockendolpher and Phillips Jr. (1990) was different to yours does not justify not discussing the results you obtained in light of previous work done on S. invicta and other invasive ants. Additionally, other studies have looked at CTmax and min for S. invicta.

For example, I found these two papers with a quick search:

M. T. Bentley, D. A. Hahn, F. M. Oi, The Thermal Breadth of Nylanderia fulva (Hymenoptera: Formicidae) Is Narrower Than That of Solenopsis invicta at Three Thermal Ramping Rates: 1.0, 0.12, and 0.06°C min −1, Environmental Entomology, Volume 45, Issue 4, August 2016, Pages 1058–1062, https://doi.org/10.1093/ee/nvw050

Clara Frasconi Wendt, Robin Verble-Pearson "Critical thermal maxima and body size positively correlate in red imported fire ants, Solenopsis invicta," The Southwestern Naturalist, 61(1), 79-83, (1 March 2016).

In the second paper, body size was found to affect CTmax which was not the case in your study and should be discussed.

The discussion also lacks references and links between paragraphs which makes the discussion confusing and your point difficult to follow. I have given some examples of these problems below and I suggest re-working this section.

L229-230 Add references

L230-232 Add references

L234 It is unclear what you mean by differing environments. More references are also needed for this paragraph. You cite the same two papers across this paragraph.

L239-243 I do not find that your examples are relevant to your study. What is the link between urban heat and your work? Why are you mentioning evolutionary change and phenotypic plasticity here?

L250-251 Give more information about which insects Janzen’s rule has been tested on and the proportion of studies which support this hypothesis.

L273 Typo, the author’s name is Bertelsmeier. I suggest checking your references for other typos. If you’re not already doing so, you may want to use a reference manager (such as Mendeley which is free and open source) to avoid future typos.

L272-274 “colder habitats” Colder than what habitats? You need to give temperature indication.

The last paragraph should be more developed. Specifically, what do your results mean

for studies looking at the future distribution of S. invicta under climate change scenario

(e.g. Bertelsmeier, Cleo, et al. "Worldwide ant invasions under climate change."

Biodiversity and conservation 24.1 (2015): 117-128).

RESPONSE: We did not suggest future distributions under climate change but

indicated that our results show that S. invicta can spread to and occupy lower

temperature habitats than currently understood and, of course, those lower

temperatures may decrease with climate change.

Ok, I do not think you should suggest future distributions either, merely that future climate change models should include the shift in CTmax and CTmin with elevation in S. invicta. Your results potentially show that more habitats are potentially at risk of invasion than previously predicted, which is important.

7. PLOS authors have the option to publish the peer review history of their article (what does this mean?). If published, this will include your full peer review and any attached files.

Reviewer #1: No

Reviewer #2: No

---

## [Author Response · Author response to Decision Letter 1]

30 Mar 2020

The authors have responded to most of my comments to my satisfaction. A few of them still need attention and I have also made some additional comments. My main concern is that the discussion lacks references regarding previous work done on the thermal tolerance of S. invicta and other invasives. As a result, I find that the significance of this study has not been appropriately discussed.

Abstract

L45 “winter dormancy” I don’t think you have mentioned winter dormancy anywhere else. Edit this sentence or explain how this term is relevant to RIFA colonies in this region in the introduction and abstract.

RESPONSE: We removed “winter dormancy” and changed the sentence to read “suggesting that greater metabolic rates were not needed to sustain these ants at high elevationswhich better ties in to our Discussion. 

Introduction

I think it would be interesting to mention the observed and potential impacts of S.

invicta in the Blue Ridge escarpment region. Is the invasion threatening important

ecosystems? Have you seen colonies penetrating undisturbed habitats?

RESPONSE: Our research was focused on regional-scale patterns in S. invicta

invasion, and hence we must be careful to not overrun our data. Moreover, we have

not observed S. invicta in undisturbed habitats. However, we added the following to the

Discussion: “Solenopsis invicta invades anthropogenically disturbed habitats (King and

Tschinkel 2008), and increased fragmentation in the Blue Ridge Escarpment region

(Turner et al. 2003) suggests that the region may experience still greater impacts of this

non-native ant. Moreover, at high elevations, the open features of sensitive rock

outcrop habitats may make them vulnerable to S. invicta invasions.”

The last two paragraphs of the discussion leave the reader with many questions instead of ending the paper nicely. See also my further comments on the discussion.

“Solenopsis invicta invades anthropogenically disturbed habitats (King and Tschinkel 2008) (…)” Add “preferably” as S. invicta has also been found to penetrate relatively undisturbed habitats.

“greater impacts” You need to describe what the current impacts are (if there are any) before mentioning “greater” impacts. I understand that you do not want to oversell your data, but is there any published evidence of RIFA impacts in the region or in similar ecosystems? If there is currently no evidence for impacts in the region, you should only mention that RIFA’s distribution may increase.

“sensitive rock outcrop habitats” I fail to see the relevance here as I do not think you have mentioned these habitats elsewhere in the paper (e.g. in the introduction). Are they undisturbed habitats? If so, and considering S. invicta preferably invades undisturbed habitats, is it likely this habitat is at risk of invasion? What makes these habitats sensitive? What organisms may be affected by a potential invasion? Have you seen S. invicta in these habitats? Perhaps it may be worth describing these habitats in the introduction where you describe study location.

The writing of these two paragraphs would also be greatly improved if it was more concise. e.g. “such as S. invicta shown here” is unnecessary.

The fact that you have not observed invicta penetrating undisturbed environments at your sites is interesting for researchers working on the effect of disturbance on the habitat suitability for invasive species. You may want to add this observation somewhere as a personal observation.

RESPONSE: We added “preferably” as suggested. We also took out the sentence describing “greater impacts” when we re-worked the discussion. We would rather not include a description of rock outcrops in the introduction since we did not collect any data from rock outcrops, and we removed the sentence about sensitive rock outcrops from the discussion so it did not seem out of place. 

We removed “such as S. invicta shown here” and the phrase “this non-native ant.” We added this sentence into the ‘Collection Sites’ section of our Methods: “All the S. invicta colony that we observed and collected were established in anthropogenically disturbed habitats, and we did not find any colonies in undisturbed habitats (personal observation).”

L 59-63: Above you state that S. invicta cannot colonize areas with temperatures below

-3.7°C but you do not give the minimal temperatures for the Blue Ridge escarpment

region and the Southern Appalachian Mountains. I was also unclear about the

difference between the two regions. Is the Blue Ridge escarpment part of the Southern

Appalachian? It would be useful to clarify for those of us who are not familiar with this

region.

RESPONSE: To give examples of temperatures of the Blue Ridge escarpment region,

we added an example from NOAA: “Within the last five years at least, S. invicta

colonies have been observed at elevations > 1220 m in the Blue Ridge Escarpment

where average low temperatures are < 4.5 °C (U.S. Climate Data 2019; Yeary-

Johnson 2014), and in fact, the average temperature for Macon County (where part of

this study was conducted) in January 2019 was 2.9 °C (NOAA 2019).”

The Blue Ridge Escarpment is a subset of the southern Appalachian mountains but

we changed the any instances of “southern Appalachian mountains” to the more

specific “Blue Ridge Escarpment” to clarify for readers who are not familiar with the

area. We also explained what this term means in the introduction: “As such, the

southern Blue Ridge Escarpment region in western North Carolina, U. S. (the zone of

abrupt change in elevation between the Blue Ridge and Piedmont physiographic

provinces with a vertical relief of 400 m to ca. 760 m)…”

Information about the minimal temperature for the region is still missing. You state that RIFA cannot invade areas for which temperatures go below -3.7°C and then that “As such, the southern Blue Ridge Escarpment region (…) was projected as unsuitable habitat for S. invicta due to the cold temperature extremes at high elevations (Korzukhin 2001).” You need to tell the reader what these cold temperatures extremes are and the elevation at which they were recorded to make this information (i.e. that RIFA cannot invade regions with min temp. below -3.7C) relevant to your study.

RESPONSE: We removed the information on average low temperatures in the region as we did not have elevational data to add to it. Instead we included a temperature extreme in Macon County where our high elevation ants were collected: “Within the last five years at least, however, S. invicta colonies have been observed at elevations > 1220 m in the Blue Ridge Escarpment where temperatures reached anomalous minima of -16.3 °C in Macon County, North Carolina, in 2019 (NOAA 2019; Yeary-Johnson 2014).”

L72: delete “much”. Even though you give more details about both study species in the

methods, I think a short sentence about A. picea’s current distribution would improve

this paragraph. I think you should also state why you chose A. picea for this work

rather than another native ant.

RESPONSE: We deleted “much” and included the suggested information in the

Introduction: “Aphaenogaster picea are native to the deciduous forests of the eastern

U. S. and are not only the most abundant ants in this region, but are important

ecologically as seed dispersers of many native understory herbs such as Sanguinaria,

Trillium, and Hepatica spp. (Ness et al. 2009; Lubertazzi 2012).”

Shouldn’t it be “one of the most abundant ants”?

RESPONSE: We changed it to “one of the most abundant ants.”

L76-77 change to “potential shorter optimal foraging time” (as you have not shown that

the optimal foraging time of S. invicta is shorter at high elevation) (…).

RESPONSE: This was changed as suggested.

This was not changed.

RESPONSE: This was accidently unchanged last time but now is changed as suggested. 

Methods

L 101-104: You need to define disturbance here and give examples.

RESPONSE: We added examples and changed the sentence to:

“Habitat disturbance such as the construction of buildings and roads or deforestation is

crucial for the establishment of many invasive species…”

An ecological definition of disturbance with more specific examples and references is still needed. It is unclear whether you are defining disturbance broadly (e.g. including storms, floods) or restrict your definition to anthropogenic disturbance.

RESPONSE: We are using Rykiel’s (1985) definition of disturbance and include this in the text: “Habitat disturbance, a perturbation that is caused by either biotic or abiotic forces (Rykiel 1985), such as the construction of buildings and roads or the removal of biomass by a natural force is crucial for the establishment….”

L149 How big were the mounds and how similar in size were they to each other? I think you should give an indication of colony size for each experiment.

RESPONSE: We collected subsamples of colonies and not full colonies since it is

difficult to collect every ant in a S. invicta colony, and therefore difficult to gauge how

big a colony is from the surface.

If you are unable to give an estimation of colony size, specify whether you only collected workers from mature colonies or whether you also included incipient colonies.

RESPONSE: We only collected mature colonies and have included this in the section “Collection Sites” of the Methods: “For study sampling, we collected from active, mature colonies that were at least ten meters from other colonies.”

L168 The ants were weighed and dry mass recorded. Lipids were removed using a Soxhlet extractor following the of Smith and Tschinkel (2009) except that we sampled whole-colony lipids rather than individual ants.

Change to “and their dry mass recorded”. A word is missing “Following the xxx of Smith (…)”

RESPONSE: We added “their” dry mass as suggested and added “protocol” so it reads “…following the protocol of Smith and Tschinkel…”

Figure 1: Add the full species name on the figure and delete this statement from the legend: “Solenopsis invicta colonies are marked in black and Aphaenogaster picea colonies are marked in white.”

RESPONSE: We added the full species names and deleted the statement from the legend as suggested.

Results

Thermal tolerance

The results need to be more descriptive by specifying the temperature for each

statement. For example, in the first paragraph, give us the average CTmax for minor

and major workers. In the second paragraph, it would good to specify for example, “the

thermal tolerance of ants found at higher elevation increased by xx°C compared to

ants at lower elevation”. Don’t use as many decimals as in line 259, one or two is

sufficient.

RESPONSE: We added average CTmax for major and minor workers as suggested

with the following statement: “The average CTmax was 46.6 °C for major workers and

46.3 °C for minor workers and the average CTmin was 6.9 °C for major workers and

6.6 °C for minor workers. Therefore, we did not separate the two size classes of worker

ants in further analyses.”

We also added an example of the CTmin difference: “On average the CTmin decreased

by 2.8 °C for S. invicta and 3.5 °C for A. picea as elevations increased.”

We respectfully disagree on the number of decimals in reporting statistical results, and

we prefer to report the standard three decimal places.

Here and elsewhere throughout the paper, you need to add a measure of variation (e.g. standard deviation) with each average (including in the introduction L59-63). I agree on the number of decimals to report for statistical results, my comment about the number of decimals was meant for the temperatures you reported L259 of the original manuscript.

RESPONSE: Here and throughout the paper we specified the temperature for each statement and included a standard error with each mean temperature reported to one decimal place. 

L198 “The CTmax for both species also decreased with elevation (SE < 0.001, t-value = -3.627, p-value < 0.001; Fig. 2B).”

RESPONSE: We added the mean temperature decrease for CTmax for both species: “Maximum temperature tolerance also decreased with elevation for both species (Fig. 3B; coef. = -0.001, SE < 0.001, t-value = -3.627, p-value < 0.001) by a mean of 0.7 ± 0.2°C for S. invicta and 0.8 ± 0.2°C for A. picea.”

L210 “Aphaenogaster picea had a lower CTmin than S. invicta (SE = 0.318, t-value = -4.150, p-value <0.001; Fig. 3A) whereas S. invicta had a higher CTmax than A. picea (SE = 0.15, t-value = -21.19,p-value < 0.001; Fig. 3B).”

Same comment as in my original review, give values for CTmin and CTmax for each species.

RESPONSE: We added this information to our results: “Aphaenogaster picea tolerated lower minimum temperatures (5.2 ± 0.2°C) than S. invicta (6.7 ± 0.1°C) [Fig. 2A; coef. = -1.321, SE = 0.318, t-value = -4.150, p-value < 0.001] whereas S. invicta tolerated higher CTmax (46.4 ± 0.1°C) than A. picea (43.2 ± 0.1°C) [Fig. 2B; coef. = -3.180, SE = 0.15, t-value = -21.19, p-value < 0.001].”

Lipid analysis

You need to add information about lipid content values along the gradient. A figure in

the supplementary material would also be useful.

RESPONSE: We have included a figure about lipid content in the supplementary

material.

You have not given lipid content values at different elevation in the main text. These values are important information regardless of your statistical results. You also did not reference your supplementary figure in the text.

RESPONSE: We added a reference to the supplementary figure in the lipid analysis section. We also included the mean lipid content values in this section: “Lipid content for low elevation S. invicta colonies was 0.042 ± 0.021 g ant-1, for mid elevation colonies it was 0.034 ± 0.022 g ant-1, and for high elevation was 0.044 ± 0.014 g ant-1.”

Discussion

Just a few more interrogations: How do your CTmin and CTmax values for S. invicta

compare to values obtained for this species in previous studies? Have other invasive

ants been found to have a shift in thermal tolerance at different elevation? The CTmin

value you found for S. invicta was about 6C. What happens to the workers when the

temperature is lower? Do they remain in the nest and is the temperature higher in the

nest? Have you determined the lethal thermal maxima and lethal thermal minima.

RESPONSE: Cokendolpher and Phillips Jr. (1990) found that S. invicta collected in

Texas had a Ctmax of 40.7 C, however the heating method was different than ours, so

we did not include comparisons. We did not determine the lethal thermal maxima and lethal thermal minima.

The significance of your results has not been fully explained in your discussion. That the methodology of Cockendolpher and Phillips Jr. (1990) was different to yours does not justify not discussing the results you obtained in light of previous work done on S. invicta and other invasive ants. Additionally, other studies have looked at CTmax and min for S. invicta.

For example, I found these two papers with a quick search:

M. T. Bentley, D. A. Hahn, F. M. Oi, The Thermal Breadth of Nylanderia fulva (Hymenoptera: Formicidae) Is Narrower Than That of Solenopsis invicta at Three Thermal Ramping Rates: 1.0, 0.12, and 0.06°C min −1, Environmental Entomology, Volume 45, Issue 4, August 2016, Pages 1058–1062, https://doi.org/10.1093/ee/nvw050

Clara Frasconi Wendt, Robin Verble-Pearson "Critical thermal maxima and body size positively correlate in red imported fire ants, Solenopsis invicta," The Southwestern Naturalist, 61(1), 79-83, (1 March 2016).

In the second paper, body size was found to affect CTmax which was not the case in your study and should be discussed.

RESPONSE: We added a paragraph to explain the significance of our results on body size and thermal tolerance: “We found the mean S. invicta CTmax (46.4 ± 0.1°C) to be similar to that found in other studies: 50.6 °C (Bentley et al. 2016) and 46.5 °C (for large workers; Wendt and Verble-Pearson 2016). The mean CTmin for S. invicta ants in our study (6.8 ± 0.1°C) was higher than that found in low-elevation ants collected by Bentley et al. (2016): 4.1 °C. The difference may lie in research methodology. Moreover, we found no appreciable difference in thermal tolerance between the smallest (minors) and largest (majors) polymorphic workers, whereas the relationship between S. invicta size and thermal tolerance are mixed for other studies (Bentley et al. 2016; Wendt and Verble-Pearson 2016). This suggests that there may be regional differences in how body size of S. invicta affects thermal tolerance, or that differences in research methodologies may affect cross-study comparisons.”

The discussion also lacks references and links between paragraphs which makes the discussion confusing and your point difficult to follow. I have given some examples of these problems below and I suggest re-working this section.

L229-230 Add references

RESPONSE: We re-worked the Discussion to give it a better flow and cohesiveness, and hopefully this revision makes it easier to follow.

L230-232 Add references

RESPONSE: We re-worked the Discussion to give it a better flow and cohesiveness, and hopefully this revision makes it easier to follow. 

L234 It is unclear what you mean by differing environments. More references are also needed for this paragraph. You cite the same two papers across this paragraph.

RESPONSE: We changed “differing environments” to “elevational and heat gradients.” We also added more references including: “Warren et al. (2020) found the opposite pattern in Brachyponera chinensis (Emery, 1895) invasions in the same study region used here. Brachyponera chinensis is a non-native ant from Asia, and has invaded the forests of North and South Carolina and Georgia. Populations found at high elevations in the Southern Appalachian Mountains do not appear to persist, and Warren et al. (2020) found that this pattern is explained by the ant’s apparent inability to acclimate to colder temperatures (i.e., its cold tolerance does not change with elevation).”

L239-243 I do not find that your examples are relevant to your study. What is the link between urban heat and your work? Why are you mentioning evolutionary change and phenotypic plasticity here?

RESPONSE: We included these examples to show that other studies have found that ants can exhibit a thermal tolerance shift along a temperature gradient including a gradient caused by a heat island effect. 

L250-251 Give more information about which insects Janzen’s rule has been tested on and the proportion of studies which support this hypothesis.

RESPONSE: We removed the reference from the discussion.

L273 Typo, the author’s name is Bertelsmeier. I suggest checking your references for other typos. If you’re not already doing so, you may want to use a reference manager (such as Mendeley which is free and open source) to avoid future typos.

RESPONSE: We corrected the typo and checked all of the in-paper references to the references at the bottom to make sure that we made no other typos. 

L272-274 “colder habitats” Colder than what habitats? You need to give temperature indication.

RESPONSE: We changed “colder” to “novel” to clarify that we mean habitats that the ants previously did not occupy. 

The last paragraph should be more developed. Specifically, what do your results mean

for studies looking at the future distribution of S. invicta under climate change scenario

(e.g. Bertelsmeier, Cleo, et al. "Worldwide ant invasions under climate change."

Biodiversity and conservation 24.1 (2015): 117-128).

RESPONSE: We did not suggest future distributions under climate change but

indicated that our results show that S. invicta can spread to and occupy lower

temperature habitats than currently understood and, of course, those lower

temperatures may decrease with climate change.

Ok, I do not think you should suggest future distributions either, merely that future climate change models should include the shift in CTmax and CTmin with elevation in S. invicta. Your results potentially show that more habitats are potentially at risk of invasion than previously predicted, which is important.

RESPONSE: We re-worked the Discussion section as suggested above, including rearranging paragraphs, including more relevant references, and removing references that were not pertinent to our study.

---

## [Editor Report · Decision Letter 2]

13 Apr 2020

Invasion and high-elevation acclimation of the red imported fire ant, Solenopsis invicta, in the southern Blue Ridge escarpment region of North America

PONE-D-19-27277R2

Dear Dr. Lytle,

We are pleased to inform you that your manuscript has been judged scientifically suitable for publication and will be formally accepted for publication once it complies with all outstanding technical requirements.

With kind regards,

Olav Rueppell

Academic Editor

PLOS ONE
---

## [Editor Report · Acceptance letter]

17 Apr 2020

PONE-D-19-27277R2 

Invasion and high-elevation acclimation of the red imported fire ant, Solenopsis invicta, in the southern Blue Ridge escarpment region of North America 

Dear Dr. Lytle:

I am pleased to inform you that your manuscript has been deemed suitable for publication in PLOS ONE. Congratulations! Your manuscript is now with our production department. 

With kind regards,

on behalf of

Dr. Olav Rueppell 

Academic Editor

PLOS ONE